# Domain adaptive learning for multi realm sentiment classification on big data

Maha Ijaz[1⊕], Naveed Anwar[1⊕], Mejdl Safran[2⊕]*, Sultan Alfarhood[2⊕], Tariq Sadad[3⊕], Imran[4⊕]

1 Department of Computer Science Faculty of Computing and Information Technology University of Gujrat, Gujrat, Pakistan, 2 Department of Computer Science, College of Computer and Information Sciences, King Saud University, Riyadh, Saudi Arabia, 3 Department of Computer Science, University of Engineering and Technology Mardan, Mardan, Pakistan, 4 Department of Biomedical Engineering, Gachon University, Incheon, Republic of Korea

⊕ These authors contributed equally to this work.
* mejdl@ksu.edu.sa

**Data Availability Statement:** We used publicly available datasets, the link of each dataset is mentioned below Movie reviews dataset available at: https://ai.stanford.edu/~amaas/data/sentiment/ Sentiment140 dataset available at: https://www.

## Abstract

Machine learning techniques that rely on textual features or sentiment lexicons can lead to erroneous sentiment analysis. These techniques are especially vulnerable to domain-related difficulties, especially when dealing in Big data. In addition, labeling is time-consuming and supervised machine learning algorithms often lack labeled data. Transfer learning can help save time and obtain high performance with fewer datasets in this field. To cope this, we used a transfer learning-based Multi-Domain Sentiment Classification (MDSC) technique. We are able to identify the sentiment polarity of text in a target domain that is unlabeled by looking at reviews in a labelled source domain. This research aims to evaluate the impact of domain adaptation and measure the extent to which transfer learning enhances sentiment analysis outcomes. We employed transfer learning models BERT, RoBERTa, ELECTRA, and ULMFiT to improve the performance in sentiment analysis. We analyzed sentiment through various transformer models and compared the performance of LSTM and CNN. The experiments are carried on five publicly available sentiment analysis datasets, namely Hotel Reviews (HR), Movie Reviews (MR), Sentiment140 Tweets (ST), Citation Sentiment Corpus (CSC), and Bioinformatics Citation Corpus (BCC), to adapt multi-target domains. The performance of numerous models employing transfer learning from diverse datasets demonstrating how various factors influence the outputs.

## Introduction

Blogs, Weibo, Social Networks and other similar sites are an important source of information for several analysts, politicians, and entrepreneurs who want to use the vast amount of text created by users to grow their business and provide continuous feedback through sentiment, opinions and comments specific subjects [1]. For example, in the travel industry, operators can identify opportunities to attract new tourists and improve services by analyzing reviews and ratings on points of interest [2]. The practice of evaluating text to predict how a person's

kaggle.com/datasets/kazanova/sentiment140
Citation Sentiment Corpus available at: https://cl.awaisathar.com/citation-sentiment-corpus/ Hotel reviews dataset available at: https://www.kaggle.com/datasets/jiashenliu/515k-hotel-reviews-data-in-europe.

**Funding:** This research is funded by the Researchers Supporting Project Number (RSPD2024R1027), King Saud University, Riyadh, Saudi Arabia.

**Competing interests:** The authors have declared that no competing interests exist.

attitude will be oriented toward an event or opinion is called sentiment classification. Text polarity is often used to assess sentiment. Sentiment refers to two respective types of thoughts, whether positive or negative, on multiple platforms where public opinion is valuable [2, 3]. Sentiment extraction is the cornerstone of sentiment classification, and a lot of research has been done. The subsequent vital phase is sentiment mining, which has recently grown quickly as a result of the rising number of global text data. Due to the tremendous growth in knowledge, finding and sharing information on a particular topic on the Internet has become an easy task. However, we are now left with a wealth of textual data that users generate when expressing opinions about various things. People are increasingly voicing their thoughts online on a number of subjects, including online product reviews, film studies, or political commentary, and literary. Therefore, it becomes crucial to consider different perspectives when interpreting people's intentions [4]. Sentiment analysis has become an significant job in Natural Language Processing (NLP) in the modern day across many fields. Online social platforms contain massive amounts of unstructured data. Sentiment analysis is employed to extract sentiments, assess their polarity, and measure the intensity of sentiment expressed in this unstructured content [5, 6]. In response to consumer feedback, for instance, online shops and food suppliers are constantly upgrading their services. The difficulty in this situation is that manually examining the feedback requires too much time and effort. The emergence of the field of NLP is the result of various breakthrough ideas and the perseverance of researchers. Research has introduced various methods and approaches for sentiment analysis. Existing sentiment analysis methods are roughly classified into three categories: sentiment dictionary-based models [7, 8], machine learning-based methods [9], and deep learning-based methods [10].

Recent years have witnessed a rise in popularity of deep learning (DL) approaches due to improvements in processing power and the volume of data that is readily accessible online. They provide automatic feature extraction as well as deeper visualization capabilities and improved performance compared to standard feature-based methods. Sentiment analysis mainly relies on supervised classification methods to estimate polarity and requires manual labeling of news sentiment to train models. Labeling is expensive, and there may not be enough training data to produce reliable results. Transfer learning [11] has been suggested as an alternative to manually creating datasets since it allows knowledge to be transferred from one neural network (the source) to another (the destination). The basic objective is to increase performance on the target domain while using a minimal quantity of training data by extracting more information from the source. Implementing a cutting-edge system that does not overfit despite the small dataset provided is another challenge. The main concept behind our solution is to reuse what has been learned, rather than starting from scratch, and save expensive computing power by adapting it in different realms. In theory, transfer learning, especially domain adaptation, will greatly reduce the time it takes to build new models. Transfer learning will revolutionize the way deep learning solutions are delivered to customers, allowing them to adapt general models to different realms in a time-and cost-optimized manner. Inspired by these recent advances, this study aims to determine whether transfer learning, especially domain adaptation, is a promising approach for NLP professionals. The chosen evaluation route was to measure the impact of domain vocabulary insertion and fine-tuning on downstream tasks. The research question is, "How does domain adaptation affect the performance of pretrained models on numerous downstream tasks?"

The main question is then broken down into sub-questions that assist us comprehend the many components that go into a thorough and comprehensive response:

1. What conditions must ULMFiT, BERT, RoBERTa and ELECTRA models meet for domain adaptation?

2. How does vocabulary affect the model's domain adaptation?

3. What are the benefits of fine-tuning the language model for the specific task?

4. To what extent has applied transfer learning improved sentiment analysis results in different realms?

In this work, we use Transformers through a novel framework, FARM, to apply transfer learning to many corpora written in English from different contexts, sources, and sample sizes. FARM supports parallel processing, making the model more computationally efficient and therefore suitable for commercial use. Tune the hyperparameters of the model to best suit the learning task. FARM's modular approach to language models and prediction heads simplifies transfer learning. Then, we employ latest deep learning model classifiers CNN and LSTM in different domains to resolve the sentiment analysis problem. Here, we present two types of comparisons of our work: the first involves transfer learning techniques such as BERT, ROBERTA, ELECTRA, and ULMFiT; the second compares different transfer learning models with deep learning algorithms to get the best performance. The major contributions of this study can be summed up as:

1. The FARM framework, with its highly modular design, is utilized by the Transformer architecture model to effectively recognize emotions across a variety of domains.

2. We test ULMFiT, BERT, ROBERTA, and ELECTRA on multiple domains including binary and multi-class datasets.

3. We compare ULMFiT, ROBERTA, ELECTRA, and BERT and demonstrate how the transformer-based approach surpasses all prior latest models and replaces them as the new latest model when the FARM framework is used with the appropriate fine-tuning.

4. Using binary and multi-class datasets, we experimented with deep learning methods including LSTM and CNN.

5. To better understand how transfer learning functions in multi-domain sentiment analysis, we did a number of experiments.

A comparison of findings demonstrates that FARM-fine-tuned transformer architecture beats latest methods for sentiment analysis across many realms. The rest of the document is structured as follows: The Related Work section analyzes the research, and the Methodology section describes the method we employed. The Results and Discussion section details the experiments and their findings, while the Conclusion section draws a conclusion.

## Related work

Sentiment analysis is sometimes referred to as polarity decision, evaluation extraction and opinion mining [12, 13]. According to previous studies on sentiment analysis come up with different categories of methods that had been applied in the past. Different learning methods that been performed sentiment analysis earlier discuss below.

### Traditional sentiment analysis methods

Traditional approaches to sentiment analysis rely on classifiers other than neural networks and do not use neural networks at all. These classifiers are usually referred to as supervised classifiers for sentiment analysis and are a component of machine learning. These methods need supervised training data to make decisions [14]. Most mutual techniques that had been used in the past are Support Vector Machine (SVM) [15–17], Maximum Entropy [18], naive

Bayes (NB) [19–21]. Explored techniques for identifying the FourSquare tips polarity by utilizing SVM, NB, and Maximum Entropy supervised learning approaches and the SentiWordNet unsupervised learning approach [22]. This study demonstrates a supervised model for the expansion of the sentiment lexicon. The researchers used a labelled dataset of tweets from Twitter to train the SVM classifier. They used attributes based on part-of-speech tags and information calculated from data streams that contained emoticons [23]. The lexicon-based type Sentiment Analysis required a dictionary of sentiments. For computing & sentimental lexicon modeling, the author uses an improved sentiment-oriented pointwise mutual information (SO-PMI) method which is a more effective technique for this kind of Sentiment Analysis [24]. This work provided a classification approach meant for the identification of ADRs for which the SVM was employed. The researchers have discussed various types of features, comprising features of sentiments, and showed that they enhanced Adverse Drug Reactions (ADR) recognition proficiency [25]. The article in [26] describes an automated sentiment analysis method, conducted through an online cancer survivor group, and compares it to a previously used method. Machine learning techniques, including Rotation Forest, Logistic Regression, Adaboost, and Random Forest were employed for classification. This study evaluated the negative impacts and related sentiments of HIV treatment. AdaBoost, SVM, Bagging, Neural Network Machine Learning Classifiers were used for analysis [27]. They presented a method in this article to expose the temporal cause and effect of sentiment changes in the Cancer Survivors Network (CSN) of the American Cancer Society. Using a machine learning classification model (Adaboost), they created a sentiment classifier, and through Probabilistic reasoning, they describe and clarify the changes between sentimental posts in a thread over time [28]. In this article, they addressed the categories of emotions and classified user comments into the emotion groups found in online medical forums. Machine learning classifiers such as Random Forest, Logistic Regression and Neural Network used for an automated multi-class classification framework [29]. This article highlighted an aspect-level approach for sentiment analysis and examined the diabetes-related topics on the Twitter and classification approach dependent on SentiWordNet scores [30]. The authors presented a news-based, sentiment-analysis-based approach to financial market forecasting. They created a predictive model using SVM and Particle Swarm Optimisation (PSO) to improve its parameters [31]. This paper comes up with query-based search value separation for negative and positive tweet classification using k-nearest neighbors (KNN) and Thresholding method [32].

## Deep learning sentiment analysis methods

Traditional methods are dependent on attributes along with Bag of words or a combination of terms with their sentiment scores. An algorithm and some manual manipulation often assign such sentiment scores to terms. Deep networks such as CNNs and Recurrent Neural Networks (RNNs) and word embedding methods such as GloVe, Word2Vec, and so forth turn over a new leaf of machine learning techniques in which feature engineering manually is not necessary. Word embedding and the deep neural network together have outscored traditional methods. Using a single convolutional layer, a neural net was introduced utilizing various widths and filters, preceded over time by a max-pooling layer. The ultimate classifier uses a layer that is entirely linked including drop-out, six datasets, especially Stanford Sentiment Treebank (SST), delineate performance [33]. A method was introduced, employing five convolutional layers, and implementing multiple temporal k max-pooling layers is a major difference in this study. This approach enables the most valuable features to be identified in a phrase, regardless of their particular position, maintaining their relative order [34]. In this study, they introduce a new deep CNN to predict sentiment polarity of short texts, employing collective knowledge

of word-level, character-level and sentence-level representations to evaluate an opinion score for a required phrase [35]. This article presents an evaluation of the use of the Convolutional Networks on character-level for text classification. Up to six convolution layers were used in these frameworks, preceded by three fully integrated classification layers along with simple max-pooling layers [36]. This research employed the very deep CNN to Natural language processing and created a new VDCNN framework for text processing tasks. The platform integrates VGGNet (Visual Geometry Group) with a deep residual network (ResNet). The authors have shown that it is possible to achieve more with 29 convolutional layers [37]. RNN is used more often in the NLP area compared to CNN, while RNN can manage sequence data and has benefits in contextual relationships, dependencies, and so forth, and that makes it more suited for processing text data. In this study, they used the LSTM for text classification tasks, LSTM can embed variable text regions. they incorporate LSTM region embedding with convolutional layers to produce maximum results, and LSTM's combined with CNN is more beneficial than the regional embedding approach [38, 39]. This study developed a fusion deep learning structure that first learned the sentiment embedding vector from the CNN and then used the Multi-Objective Optimisation (MOO) model to produce a set of optimisation features, and last used SVM to classify the optimised vector sentiment. They evaluate their presented method at the level of both the sentence and the aspect. It was the first effort to introduce this method of deep learning to the sentiment analysis in languages through fewer resources [40]. This article presents a dividing-and-conquer method that first categorizes sentences into different types and then analyzes sentiments independently on the base of sentences category using BiLSTM-CRF and 1d-CNN [41]. In this article, they presented a model focused on the integration of multiple attention mechanisms and recurrent neural network (RNN) to detect the sentiment of opinions. The research results indicate that on various data sets this approach performs better [42]. Interpreting the sentiments regarding medical conditions posted on social media by users by creating a deep CNN for sentiment analysis [43]. In another research, the neural model Hierarchical Tweet Representation and Multi-head Self-Attention (HTR-MSA) is introduced. The model is constructed with word and character embedding, CNN, LSTM, and multi-head attention for uncovering of drug names (DNs) and adverse drug reactions (ADRs) mentioned in tweets [44]. This article presents an established study of classification utilizing deep learning models on sentiment analysis and presents comparative outcomes of numerous deep learning networks. The dataset pre-processed through Word2Vec, and then corresponding word embedding employed. As a benchmark Multilayer Perceptron (MLP) was created. CNN, LSTM, RNN and a hybrid model of LSTM and CNN were built and implemented on the IMDB dataset. The results indicate that the hybrid model CNN LSTM perform better than the MLP and the distinctive networks CNN and LSTM [45].

## Transfer learning sentiment analysis methods

In this research, the Transfer learning approach is leverage through an unlabeled dataset from Twitter to Turkish political columns. The proposed study collected and transferred features extracted from Twitter and using them in labeled data for the Sentiment Analysis task. They also reported an improvement in classification performance [46]. This article developed a DeepMoji model using the transfer learning approach to identify sentiments, emotions, and sarcasm. So far, they have demonstrated how emojis' widespread use in social media communications can also be used to pre-train models, allowing them to pick up on emotional expression from textual data and transmit it to the target domain. Using a single pre- trained construct, they employ 8 datasets from 5 domains. They have shown that this approach will work well and is capable of achieving better outcomes that are comparable to other methods

[47]. In another article, the authors presented a multi-Task Learning-based bidirectional transformer architecture for topic and sentiment classification tasks. They simultaneously trained on a variety of datasets, including AG News, Stanford Sentiment Treebank (SST-2), and Movie Review (MR), and showed gradual advancements over current designs [48]. A transfer learning strategy for sentiment analysis employing a sizable dataset is presented by the authors in another publication. They used a pre-trained model to determine the tweet's sentiment intensity level among seven categories after training a model to predict if a Twitter post is good, negative, or neutral. The application of transfer learning and BiLSTM led to improved outcomes [49]. In this paper, the researchers introduce the Universal Language Model Fine-tuning (ULMFiT) model as a useful transfer learning method for carrying out NLP tasks. By utilising a sizable Wikipedia corpus, they created a language model, which they then improved using the IMDB dataset. They showed that in both transfer learning and modern text categorization tasks, this methodology beat all previously introduced strategies [50]. In this article, they presented a ULMFiT based transfer learning technique method for the Sentiment Analysis task. a pre-trained model built using Wikipedia and then fine-tuned the language model independently on two datasets and the Transfer Learning model outperformed in both datasets [51]. In this work they experimentally compare hierarchical models and transfer learning for classification of sentiments at the document level. They demonstrate that non-trivial hierarchical models outperform subsequent benchmarks and transfer learning in five languages on document-level sentiment classification [51].

(1) Traditional transfer learning using the Manifold Dynamic Distribution Adaptation (MDDA).

(2) Deep transfer learning using the Dynamic Distribution Adaptation Network (DDAN).

Experiments indicate that MDDA and DDAN improve transfer learning [51]. In this research [52] they propose a probabilistic generative model of words that enable them to handle multi-source and multi- target domains. They have employed the transfer learning approach referred to the situation where what has learned in one setting (source domain) is exploited to improve generalization to another setting. Document level polarity classification tasks have performed by employing multi-source and target domain data. This model enables them to extract words to create domain-dependent word polarity dictionaries. They use the Gibbs Sampling method for learning parameter values. They performed two trials:

1. Fix the number of targets domain and increase the number of source domain.

2. Fixing the number of source domain, increase the number of target domain.

Datasets: they used 17 domains and 10000 documents from Multi-Domain Sentiment Dataset.

Unlabeled data was used in the beginning to estimate word-level or phrase-level statistics, which were later included in supervised model-like features. Unlabeled data was observed and allowed for broad adoption in recent techniques than word-level data. Text for various target tasks was encoded into appropriate vector representations using phrase-level or sentence-level embedding, which could be learned from an unlabeled corpus. The more literature of the previous models can be found in Table 1.

It is challenging to construct a model from scratch owing to time restrictions or computing limits, pre-trained models with broad potential and possibilities are presented. They serve as a baseline against which to either refine an existing model or compare a newly generated model. Language models that have been pre-trained and are frequently loaded with generic corpora such as Wikipedia The distribution of the target domain corpus, on the other hand, is usually

**Table 1. List of previously applied transfer learning models for sentiment analysis.**

| Literature | Technique | Dataset | Accuracy % |
|---|---|---|---|
| [53] | LSTM | IMDB | 91.80% |
| | | SST- 5 | 53.70% |
| [54] | biLM (bidirectional LSTM) | SST- 5 | 54.70% |
| [55] | DC-MCNN | SST | 86.99% |
| | | AFF | 90.08% |
| | | SE16-T5 | 85.02% |
| [50] | AWD-LSTM | IMDB | 95.40% |
| [56] | CNN_FT | Amazon reviews | 85.35% |
| [57] | ART (aligned recurrent transfer) | Amazon reviews | 85.80% |

distinct (in our case, a movie or product review). As a result, prior to training each classifier, we spend a few iterations fine-tuning the source language model on the target corpus (without labels). The semi-supervised learning group is the primary focus of our research on natural language processing. Finding a good starting point rather than modifying the supervised learning subject is the aim of unsupervised pre-training, a sort of semi-supervised learning. In NLP, transfer learning is presently commonly done as a multi-step procedure in which a complete network is initially unsupervised pre-trained with a language modeling purpose. The model is then fine-tuned on a new task using a supervised technique (with some labeled data) before being employed for tasks like text categorization. In order to study a semi-supervised approach to language understanding issues, we combined unsupervised pre-training and supervised fine-tuning. We would acquire a general representation that requires little modification and can be applied to a variety of tasks. We presume that we have access to a large, unlabeled text corpus as well as numerous datasets with manually annotated training examples (target tasks). The study's key contribution is the application of transfer learning to the multi-domain Sentiment categorization task using a fine-tuning approach and deep learning architecture.

## Methodology

Based on the history, literature study, and state of the art from the previously-mentioned two sections, this section delves deeper into the techniques adopted in this paper. To be precise, we intend to perform Sentiment Analysis leveraging Natural Language Processing (NLP) technologies. The sentiment analysis ought to polarize the data set by identifying positive, negative, and neutral sentiment. The proposed methodology is presented in Fig 1 and details description is found in Fig 2.

### Dataset

In the relatively young field of sentiment analysis, enormous amounts of data are analysed to provide insightful information about a particular topic. Sentiment analysis has drawn a lot of attention as a result of the wide accessibility of review websites like CNET2, IMDB, and Rotten Tomatoes4, as well as the quick adoption of social media sites like Facebook, Twitter, and Reddit. In this study, we conduct experiments and assess how well different Transfer Learning approaches combined with deep learning techniques address the issue of sparse annotated data by using less annotated datasets from the multiple-domain.

We ran trials on five distinct domain datasets.

**Hotel reviews.** We used the hotel reviews dataset gathered from the Booking.com website, and all of the information in the file is already publicly available. Booking.com owns the data.

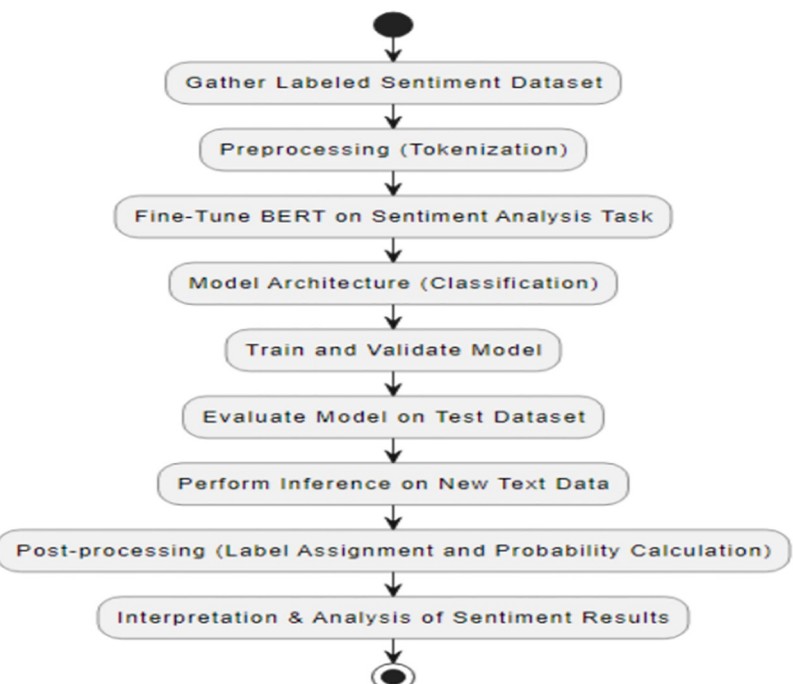

**Fig 1. Proposed steps.**

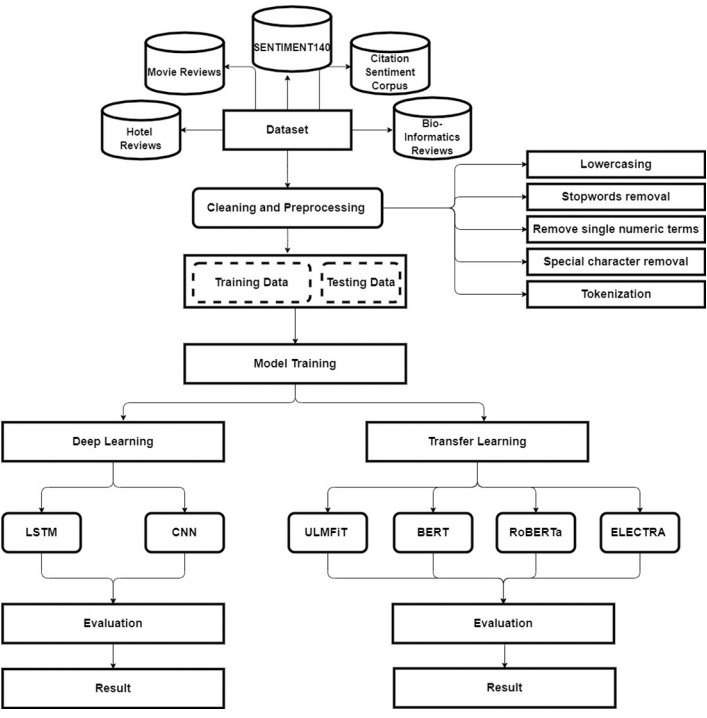

**Fig 2. Methodology.**

This database comprises 20,000 customer reviews marked as negative and positive. The dataset is unbalanced, and the majority of the response is positive, which implies that people are generally satisfied with the service. We conducted tests by choosing K instances and producing training datasets of 7, 10, 15, and 20K to explore the effects of variations in dataset size (see Table 2) [58].

**Movie reviews.** We used the dataset of movie reviews obtained from the IMDb website compiled by Andrew Maas [59]. The dataset is freely accessible to the public, and we took 20,000 movie reviews from the dataset that classified as positive or negative sentiment. The dataset is evenly split between positive and negative entries, making it well-balanced. We carried out additional experiments to evaluate the consequences of the dataset adjustments. 7K, 10K, 15K, and 20K training datasets were produced overall [60] (see Table 3).

**Sentiment140 tweets.** Sentiment classification on social networking sites is challenging, specifically on microblog textual content based on short status posts, flexible words and emoji's, a mixing of several text representations and patterns. We used the dataset of tweets *Sentiment140* from Stanford University. This dataset contains 20,000 tweets obtained through the Twitter API, a binary-class dataset with positive and negative sentiment target classes. The dataset is balanced, with positive and negative records. We have conducted additional experiments to test the effects of modification in only datasets sizes. The formation data is set to 7K, 10K, 15K and 20K [61] (see Table 4).

**Citation sentiment corpus (CSC).** When it comes to exploiting publicly accessible high-quality datasets for citation sentiment analysis, data is rare. The class distribution of Athar's Citation Sentiment Corpus, also known as CSC, is extremely uneven. The dataset has a total of 8,736 instances and is divided into three categories: target classes for positive, negative, and neutral sentiment (see Table 5). 756 records were eliminated during the dataset cleaning

**Table 2. Hotel reviews data statistics.**

| Labels of Reviews | Total Number of samples | Train samples | Test samples |
|---|---|---|---|
| Positive/Negative | 7000 | 6000 | 1000 |
| Positive/Negative | 10000 | 8000 | 2000 |
| Positive/Negative | 15000 | 12000 | 3000 |
| Positive/Negative | 20000 | 16000 | 4000 |

**Table 3. Movie reviews data statistics.**

| Labels of Reviews | Total Number of samples | Train samples | Test samples |
|---|---|---|---|
| Positive/Negative | 7000 | 6000 | 1000 |
| Positive/Negative | 10000 | 8000 | 2000 |
| Positive/Negative | 15000 | 12000 | 3000 |
| Positive/Negative | 20000 | 16000 | 4000 |

**Table 4. Sentiment140 tweet statistics.**

| Labels of Reviews | Total Number of samples | Train samples | Test samples |
|---|---|---|---|
| Positive/Negative | 7000 | 6000 | 1000 |
| Positive/Negative | 10000 | 8000 | 2000 |
| Positive/Negative | 15000 | 12000 | 3000 |
| Positive/Negative | 20000 | 16000 | 4000 |

**Table 5. Citation sentiment corpus statistics.**

| Positive | Negative | Neutral |
|---|---|---|
| 829 | 280 | 7627 |
| 9.49% | 3.21% | 87.30% |

**Table 6. Bioinformatics Citation Corpus (BCC) statistics.**

| Positive | Negative | Neutral |
|---|---|---|
| 1984 | 959 | 1180 |
| 48.12% | 48.12% | 23.26% |

process. These records were removed because they either duplicated content that already existed or gave different [62].

**Bioinformatics citation corpus (BCC).** For the bioinformatics citation corpus, we used a collection of clinical trial articles that included 4,123 annotated citations (neutral, negative and positive) as presented in Table 6. The class distribution of this dataset is unbalanced. We isolated 20% of the reviews from the training set for validation [63].

## Research design

This section describes the methodology employed in this paper based on the premise of introduction, literature review, and state-of-the-art presented in the previous two chapters. More specifically, sentiment analysis leveraging NLP methods is a cornerstone of the workflow. This section discusses preprocessing techniques as well as the evaluation of different classification techniques.

When directed towards a specific domain, all sentiment-analysis method works fine. However, when domain borders have exceeded, they experience substantial performance loss. In terms of sentiment analysis, writing style has an influence, and It referred to as a domain boundary. For instance, the expression "*Nice perfume. Must you marinate in it?*" contains words that express a positive sentiment, yet it has to label as a negative sentiment line of text. This article aims to evaluate the efficacy of various transfer learning approaches in addressing the challenge of sparse annotated data. By utilizing a limited amount of annotated datasets in a multi-domain setting, we intend to demonstrate the potential of these approaches.

**Data cleaning and preprocessing.** In most situations, data selection is the initial stage because raw data is often insufficient. The first and most essential stage is data preparation. A good outcome can be achievable from well-executed data preprocessing and vice versa. For a variety of reasons, raw data might be disorganized and confusing. Some characteristics are lost because they include incorrect values or duplicate data. Data selection can minimize data redundancy by a substantial amount.

*Data attributes selection*. A single record can include a variety of data, according to Hotel Reviews Dataset, comprises properties such as "Hotel Address," "Review Date," "Hotel Name," "Negative Review," "Positive Review," "Reviewer Nationality," "Total Number of Reviews," "lat", "lng" "days since the review," "Tags," and so forth. The relevant qualities for subsequent usage of this paper are "Negative Review" and "Positive Review". All the other attributes can be removed in order to decrease the duplication of the dataset.

*Integrity check*. After data selection, the data is still not "clean" and not suitable to process. Performing an integrity check entails removing all incorrect data. First, check the data type to

ensure that it is accurate and consistent. Second, then check for any null records that may exist. Finally, it is necessary to determine whether there are any outliers. For instance, check If the score column has an int64 type, or see if there are any insertions with no text attribute at all. Delete the incorrect data or change it to the proper data if necessary.

*Deduplication*. There are often duplicate values in datasets, whether those are due to a variety of technical reasons or human-caused errors. Duplication increases data redundancy, which can lead to statistical bias and negatively impact experiment findings.

*Data preparation*. Text processing is imperative for the efficacy of the analysis as textual information contains a lot of noisy and unstructured text.

*spaCy and NLTK*. Spacy is a Python open-source toolkit for natural language processing (NLP) activities that allows implementations for textual preprocessing, deep learning, and several other tasks in the realm of NLP. It parses and analyses enormous amounts of text. It tackles NLP tasks by implementing standard methods in the most efficient way possible. Natural Language Toolkit (NLTK), in the same vein as Spacy, with user-friendly interfaces for over 50 corpora and lexical resources, such as WordNet. It contains several text processing modules, such as tokenization and stemming, and classification. For text preparation, the NLTK package was employed for tokenization, lemmatization, and stop word removal.

*Tokenization*. In NLP, tokenization has a significant impact on the rest of your system. Tokenization is the process of separating the raw text into small parts as illustrated in Fig 3. Unstructured data and natural language text are broken down into distinct parts using a tokenizer. When a phrase, paragraph, or full-text document gets tokenized, it has broken down into smaller groupings, such as single words or phrases that mark as a token.

The most common way of creating it is through whitespace, assuming that whitespace worked as a delimiter. It helps in understanding the context or building a model for the NLP. By examining the word order, tokenization assists in deciphering its content by analyzing how words are. The occurrences of tokens in a document can be employed explicitly as a vector representing that content.

*Stopwords removal*. In NLP, to minimize the raw text input, stopwords are a considerable step. Most languages have stop words (e.g., modal verbs, pronouns, etc.). Every language has common words that emerge more frequently than others, and Natural Language Processing tools provide a list of stopwords. Stopwords are the most prevalent words in any natural language. This, the, and an are all examples of stopwords in English. These stopwords may not add much value to the meaning of the text when it comes to analyzing textual data and building NLP models. Stop word removal should utilize with discretion in sentiment analysis. Terms such as "not" and "none" can flip the polarity of sentences from negative to positive. Consider the following text string: "There is a book on the desk." While parsing the sentence, the terms "is," "a," "on," and "the" have no meaning as illustrated in Fig 4. Words as "there," "book," and "desk" are keywords that indicate what the sentence seems to be about.

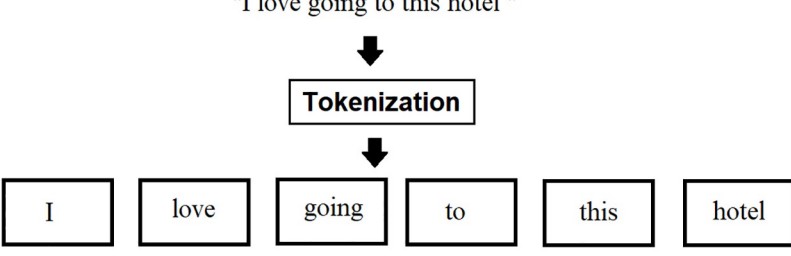

**Fig 3. Tokenization.**

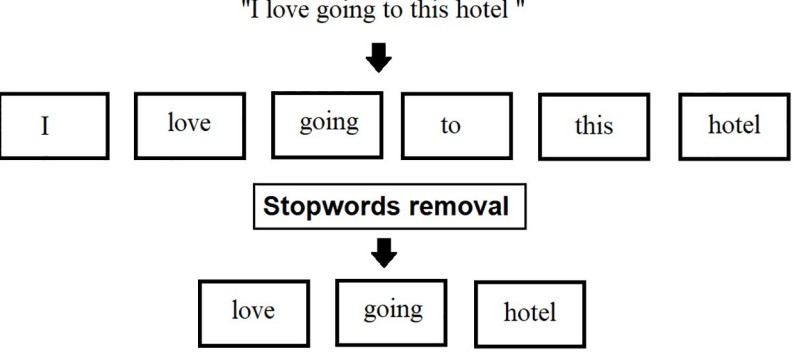

**Fig 4. Stopwords removal.**

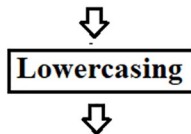

**Fig 5. Lowercasing.**

*Lowercasing.* It is the most basic preprocessing approach, which involves lowercasing every single character in the incoming text. Lowercasing is a common approach in deep learning library modules and word embedding packages owing to its simplicity. Although its beneficial characteristic of lowering sparsity and vocabulary size. Lowercasing may possess a detrimental influence on system performance by raising ambiguity.

For instance, "*Apple is asking its manufacturers to move MacBook Air production to the United States.*" after lowercasing would be like "*Apple is asking its manufacturers to move macbook air production to the United States.*" In this hypothetical scenario, the apple firm and the apple fruit would deem to be identical entities as shown in Fig 5.

*Special character and single numeric term removal.* Non-alphanumeric characters refer to as Special Characters (SC) and numeric terms. These characters commonly contain reviews, references, figures, and so forth. No emotion conveyed by Special Characters (SC) and numbers. Thus eliminating them could help in combining multiple terms that previously unrelated terms. These characters do not contribute to text understanding and create noise in algorithms.

## Fine-tuning with farm framework

Transfer learning involves adapting knowledge gained from one task to another. The FARM Framework for Adapting Representation Models, with its modular design, simplifies the adoption of transfer learning for NLP. The given modules enable the user to do fine-tuning for downstream tasks conveniently, observe the training, and finally assess the model result. The

structure of every module is rendered as precise as feasible, and the building blocks omit the majority of the standard implementation overhead. In many situations, applying a language model to your particular NLP challenge necessitates extensive preprocessing. FARM's modular structure makes preprocessing far more efficient and can be easily customized. Processors are used to converting input files into PyTorch datasets. In this research, FARM is used to fine-tune BERT for predicting sentiment from textual data. It is created with transformers and possesses a modular structure for the language models and prediction heads.

**Data handling.**   Especially likening to BERT wordpiece tokenization, which employs the typical Hugging Face technique, the FARM modular structure makes preprocessing far more straightforward and adaptable. Data handling is crucial in any deep learning setting. The framework accepts input files and provides processing-ready datasets that could be utilized for additional downstream operations like training, fine-tuning, or conducting inference. Processors are used to converting input files into PyTorch datasets. The processor requires a tokenizer for this purpose, which could also be loaded based on the language model. Data Silo controls the provided dataset and sends it into the Dataloader, which involves parallelization to speed up data processing. After reading and formatting the data, the Datasilo sends it to the related module based on the task.

**Modelling.**   FARM offers a broad and adaptable technique to transfer learning. The modeling in the FARM framework is formed on the idea of Adaptive Models. FARM's adaptive model serves as the foundation for end-to-end transfer learning. It includes the language model and the prediction heads as its two primary parts.

1. Language model: Tokens are transformed into vector representations using pre-trained language models, including BERT or XLNet [64]. As previously stated, the pre-trained language model in our situation is $BERT_{base}$, RoBERTa, and ELECTRA.

2. Prediction head: A prediction head is a layer added to the language model to represent a specific downstream task. Vector representations from the language model are given into the prediction head, which generates predictions for the downstream task. Based on a pre-trained Language Model, this parent class inserts one or more Prediction Heads for a given downstream job. It runs forward passes, figures out the neural network's overall loss, and then uses the result to back-propagate the weights of the entire Adaptive Model. Many combinations can be rapidly executed using this building block technique.

**Training.**   The user must first configure the data handlers and the Adaptive Model in order to start training a new model. The training procedure, which may be used as a method within the Adaptive Model class, should then be executed after they have specified the hyperparameters. On the development set, frequent evaluations were carried out throughout the training. After the procedure is finished, the performance on the validation (or test) set is evaluated once more, and a new Adaptive Model object with adjusted weights is returned.

**Hyperparameters.**   The hyperparameters can be adjusted to produce the best outcomes. These variables need to be manually adjusted because the model itself cannot learn them. The term "hyperparameters" refers to a wide range of variables, such as the learning rate, training batch size, dropout rate, number of training epochs, and any other optimizer-related variables that can be altered, like the Adam optimisation technique. We employ bert-base-cased for BERT [48], roberta-base for RoBERTa [65], and google/electra-base-discriminator for ELECTRA [66] as baselines for deep neural network techniques.

We use FARM to build an adaptive model with a classification prediction head to fine-tune for this purpose. By using weightings through CrossEntropy, FARM classifiers successfully

**Table 7. Hyperparameters employed for Bert, Roberta and Electra.**

| Task | Model | Learning rate | Batch-size | Epochs | MSL |
|---|---|---|---|---|---|
| HOTEL REVIEW | $BERT_{base}$ | $3e-5$ | 10 | 5 | 512 |
| | $RoBERTa_{base}$ | | | | |
| | $ELECTRA_{base}$ | | | | |
| MOVIE REVIEW | $BERT_{base}$ | $3e-5$ | 10 | 5 | 512 |
| | $RoBERTa_{base}$ | | | | |
| | $ELECTRA_{base}$ | | | | |
| SENTIMENT140 | $BERT_{base}$ | $3e-5, 2e-5, 1e-5$ | 10 | 5 | 512 |
| | $RoBERTa_{base}$ | | | | |
| | $ELECTRA_{base}$ | | | | |
| CITATION SENTIMENT CORPUS(CSC) | $BERT_{base}$ | $1e-5$ | 5 | 5 | 512 |
| | $RoBERTa_{base}$ | | | | |
| | $ELECTRA_{base}$ | | 8 | 8 | |
| BIOINFORMATICS CITATION CORPUS (BCC) | $BERT_{base}$ | $2e-5$ | 9 | 9 | 512 |
| | $RoBERTa_{base}$ | $1e-5$ | 6 | 6 | |
| | $ELECTRA_{base}$ | $3e-5$ | 8 | 8 | |

handle unbalanced classes. A straightforward feed-forward network is used in the newly added output layer to calculate prediction probabilities using Softmax. We used the following hyperparameters for the fine-tuning procedure: a maximum sequence length of 512 tokens, 5 iterations, a batch size of 10, and a learning rate of $3e-5$. Table 7 summarizes information on the fine-tuning hyperparameters that we employ for BERT, RoBERTa, and ELECTRA.

$$(MSL = Maximum\ Sequence\ Length)$$

**Hyperparameters of universal language model.** We are concerned about a model that could do a wide range of tasks with efficiency. With this intention, we employed the same set of hyperparameters across works. We used the ULMFiT authors Howard and Ruder's AWD-LSTM language model, which had a 400 embedding size, three LSTM layers, and 1152 hidden activations per layer. Using Backpropagation Through Time (BPTT) and a batch size of 70, the model was trained. A hidden layer with a size of 50 made up the classifier. With $\beta1 = 0.9$ and $\beta2 = 0.99$, we used the Adam optimisation approach. We utilised a dropout multiplicity of 0.3 and a batch size of 32 for language model fine-tuning. The learning rate was adjusted to 0.001 for fine-tuning the language model and to 0.015 for the classifier.

## Deep learning experiment

Keras [67], a deep learning toolkit, includes an easy-to-use import method for acquiring these preprocessed reviews. By mapping reviews to a dictionary that contains the D most frequently occurring terms in the dataset, where D is a number, the function transforms reviews into a series of word indices. According to word frequency, the indices are assigned, with index 2 designating the second-most frequent term in the dataset. Unknown terms that are absent from dictionaries are placed in index zero. They are terminated for longer reviews, while shorter reviews are padded with zeros to match a predetermined maximum sequence length. Ultimately, these indices are converted into E-dimensional embeddings by the neural network's first layer. For this experiment, the maximum padded sequence length was 2000 words.

**Embedding.** The network is divided into numerous branches, each having integrated kernels for data input and convolution. 128 embeddings are used during the network's training and testing processes. The input reviews are initially supplied into the network's first layer as a series of indices. The first layer encapsulates each word in a 'e'-sized vector of a particular length. For instance, if we use an embedding size of 32 and we have a vector of 300 word indices, the resulting vector will have 300 individual vectors, each of which will have a length of 32. The embedding layer is a matrix of trainable weights in and of itself. To create vectors corresponding to each word index, it multiplies matrices. In order to improve each word's representation in the network during training, the embedding layer continuously improves its embeddings.

**Convolution.** Each of the b branches retrieves the output of the embedding layer. A 1-dimensional convolution layer with a distinct kernel size c is the first layer on each branch. In 1-dimensional convolution, the kernel is formatted as kernel size by embedding size (c x e). Convolution is applied to every word, and various outputs are produced using various filters f. With a kernel size of c, the CNN layer's job is to collect word combinations. This makes it possible to comprehend how words interact when they are used in combination. In the case of c = 3, the layer, for instance, recognises three words at once, resulting in the formation of three-word pairings. The input height of this layer, which is affected by the filters, determines the output shape (words by f).

**Activation.** Each branch processes the CNN layer output using a *ReLU* activation function, replacing all negative outputs with zero. The *ReLU* layer's goal is to give the network some nonlinearity. This layer's output keeps the same shape as its input.

**Max pooling.** Following *ReLU* activation, each network branch goes through a 1-dimensional max pooling operation, which condenses the input by choosing the highest value within each kernel size. By using this method, the input's size is decreased without losing any of its key components. The main goal of this layer is to avoid overfitting while preserving the ability for subsequent processing. The 1-dimensional max-pooling kernel's shape is intended to correspond to the data's width, much like the CNN layer does. As a result, the parameter 'p' provides the kernel's width and denotes a kernel shape of 'p' by the data width. This method makes sure that pooling is done while taking the nature of the word-based data into account. The height of the output of this layer therefore decreases and is influenced by the kernel size 'p' (*inputheight* $\div$ *p*).

**Dropout.** In this layer, a predetermined percentage of the input values are arbitrarily set to 0. This approach is used to a subset of the insights (d). The dropout layer reduces overfitting by allowing the network to be more generalist and less focused on particular input components. The input and output shapes are the same.

**Long short-term memory (LSTM).** The model includes an LSTM layer with a specified number of units, 1. The LSTM is chosen for its ability to handle sequential data, where previous inputs can influence subsequent outputs due to the layer's persistence. The output length is equal to one unit.

**Dense.** The final layer is a completely linked layer that outputs a single value from the input. A simplified *sigmoid* activation function is then used to adjust the output between 0 and 1. In the end, there is only one output.

**Loss function and optimizer.** A binary cross-entropy loss function is used to calculate the loss in two classes, namely 0 and 1, throughout the network's construction. In this study, a score of 0 signifies negative emotion while a score of 1 indicates positive emotion. The loss is computed using the single and final outputs of the dense layer. For multi-class, the categorical_crossentropy function is used for compilation. This function estimates the loss in three classes (0,1 and 2). For this study, 0 denotes negative, 1 indicates neutral and 2 intimates

positive emotion. The architecture is also built using an optimizer; the RMSprop, Adam and Stochastic Gradient Descent (SGD) were employed in the tests. Each optimizer utilized different learning rates and decay variables.

**CNN model.**   The first CNN layer manipulates 32 filters to process the word embedding matrix, while the next layer employs MaxPooling1D of size 2. The flattened vectors were transmitted to a dense layer of 10 neurons. Finally, the dense layer's output is being sent to a sigmoid layer, which calculates the class probability. Because the performance of deep neural networks is highly dependent on the hyper-parameters chosen, we experimented with batch sizes and epochs.

**LSTM model.**   This architecture includes a tokenizer, embedding layer, SpatialDropout1D layer, LSTM layer, and dense layer. The input is passed through a tokenizer, which converts each token into an integer value based on the vocabulary index. After that, they're converted into representative vectors using an embedding layer. We use SpatialDropout1D to decrease the parameters after each train to reduce overfitting. To extract abstract characteristics, we use a LSTM layer built-up of 100 LSTMs with recurrent_dropout of 0.2. After normalizing in the dense layer, the labels of the sub-tasks are acquired.

*Technical resources.* For training model, we used Nvidia K80 GPU with 12 GB GDDR5 VRAM, Intel(R) Xeon(R) processor with two cores @ 2.30GHz, and 12.6 GB RAM. We used Pytorch to implemented our model deep learning framework with leveraged Fastai v1 libraries to fine-tune the LM.

## Evaluation

Training, Testing accuracies are primarily employed in the study to evaluate the various Transformer-based pre-trained models and deep learning models. Furthermore, F1 scores were utilized to evaluate the model performance while applied to balanced and unbalanced data.

**Accuracy.**   The accuracy of a method is a measure of its correctness. It is generally computed as the proportion of the number of right categories to the total number of classifications. Training accuracy is frequently employed to evaluate the model's performance after a single training epoch. Test accuracy, on the other hand is the model's accuracy once it has been fully trained. Model accuracy is not the foremost assessment metric since the data is skewed. The accuracy measure can be considered as the prominent assessment metric for the balanced.

$$Accuracy = \frac{TP + TN}{TP + FP + TN + FN} \tag{1}$$

Where TP, TN, FP, and FN stand for True Positives, True Negatives and False Positives respectively.

**F1-score.**   The precision P and recall R of the test are considered to compute the F1-score. Eq 2 gives the formula, where TP is true positives, FP is false positives, and FN is false negatives. F1 is the harmonic mean of precision and recall; while both precision and recall are flawless, F1 yields 1.

$$F1 = \frac{2PR}{P + R} = \frac{2TP}{2TP + FP + FN} \tag{2}$$

A macro-average F1 score calculates the metric separately for each class and then averages it, thereby treating all classes equally.

$$F_1^M = \sum_{I=1}^{n} w_i F_{1_i} \tag{3}$$

A micro-average F1 score computes the average metric by aggregating the contributions of all classes. Micro-average is preferred in a multi-class classification setup to decline class imbalance:

$$F_1^\mu = \frac{2 \sum_{k=1}^{n} TP_k}{2 \sum_{k=1}^{n} FP_k + \sum_{k=1}^{n} TP_k + \sum_{k=1}^{n} FN_k} \tag{4}$$

The True Negatives determine a model's reliability, which is frequently measured by the F1 score. A balanced score such as F1 considers the weighted average of Precision and Recall when evaluating a model's performance when comprehending both False Negatives and False Positives of a model bears a financial penalty. When the distribution of target labels in the data is uneven, the F1 score is especially useful.

## Results and discussion

The findings for the deep learning and transfer learning approaches used in this study are presented in this part. For evaluation, we consider accuracy and F1-score as our key performance indicators. Tables 8 and 9 show the results of the models with transfer learning and deep learning. We found that the models with transfer learning performed significantly better than the deep learning models. We conducted investigations to see how different dataset sizes affected the results. We generated training data by selecting K instances of sizes 7K, 10K, 15K, and 20K. The Transfer Learning model works well across a dataset of different domain knowledge and achieves better accuracy. For sentiment classification of five distinct datasets, the researchers used four transfer learning techniques anointed (ULMFIT, BERT, ROBERTA, and ELECTRA) and two Deep Learning techniques named (LSTM, CNN). The results are reported in the tables below.

Table 8 presents a comparison of the transfer learning method's performance on five datasets: the hotel review dataset, movie review dataset, SENTIMENT140, Citation Sentiment Corpus (CSC), and Bioinformatics Citation Corpus (BCC). The table summarizes the findings of the transfer learning technique on these datasets. The accuracy measure was utilized as the primary assessment metric. Other assessment measures, such as (F1-score), will be evaluated along with it. Other experiments have been performed to examine the influence of modifying

**Table 8. Results of applied transfer learning models for sentiment analysis.**

| Datasets | Size | ULMFIT | | BERT | | ROBERTA | | ELECTRA | |
|---|---|---|---|---|---|---|---|---|---|
| | | A | F1 | A | F1 | A | F1 | A | F1 |
| Hotel review | 7K | 0.928 | 0.927 | 0.941 | 0.941 | 0.946 | 0.945 | 0.918 | 0.917 |
| | 10K | 0.940 | 0.939 | 0.942 | 0.942 | 0.946 | 0.946 | 0.918 | 0.917 |
| | 15K | 0.941 | 0.941 | 0.948 | 0.948 | 0.950 | 0.949 | 0.924 | 0.923 |
| | 20K | 0.952 | 0.951 | 0.947 | 0.947 | 0.955 | 0.954 | 0.931 | 0.930 |
| Movie Review | 7K | 0.921 | 0.921 | 0.917 | 0.917 | 0.936 | 0.936 | 0.944 | 0.944 |
| | 10K | 0.921 | 0.921 | 0.908 | 0.908 | 0.941 | 0.941 | 0.936 | 0.936 |
| | 15K | 0.926 | 0.926 | 0.918 | 0.918 | 0.935 | 0.935 | 0.938 | 0.938 |
| | 20K | 0.931 | 0.931 | 0.911 | 0.911 | 0.940 | 0.940 | 0.941 | 0.941 |
| Sentiment140 | 7K | 0.754 | 0.754 | 0.804 | 0.804 | 0.836 | 0.836 | 0.795 | 0.795 |
| | 10K | 0.770 | 0.770 | 0.805 | 0.804 | 0.823 | 0.823 | 0.801 | 0.800 |
| | 15K | 0.770 | 0.770 | 0.810 | 0.810 | 0.834 | 0.834 | 0.801 | 0.801 |
| | 20K | 0.780 | 0.780 | 0.814 | 0.814 | 0.836 | 0.836 | 0.812 | 0.812 |
| CSC | 8K | 0.910 | 0.897 | 0.938 | 0.902 | 0.937 | 0.902 | 0.932 | 0.898 |
| BCC | 4K | 0.855 | 0.853 | 0.920 | 0.880 | 0.894 | 0.840 | 0.885 | 0.829 |

**Table 9. Results of applied transfer learning models for sentiment analysis.**

| DATASETS | SIZE | LSTM | | CNN | |
|---|---|---|---|---|---|
| | | **A** | **F1** | **A** | **F1** |
| **HOTEL REVIEW** | **7K** | 0.924 | 0.924 | 0.901 | 0.901 |
| | **10K** | 0.925 | 0.924 | 0.898 | 0.897 |
| | **15K** | 0.925 | 0.924 | 0.898 | 0.897 |
| | **20K** | 0.941 | 0.940 | 0.904 | 0.903 |
| **MOVIE REVIEW** | **7K** | 0.801 | 0.801 | 0.839 | 0.839 |
| | **10K** | 0.820 | 0.820 | 0.840 | 0.840 |
| | **15K** | 0.862 | 0.862 | 0.857 | 0.856 |
| | **20K** | 0.885 | 0.885 | 0.867 | 0.866 |
| **SENTIMENT140** | **7K** | 0.688 | 0.690 | 0.656 | 0.654 |
| | **10K** | 0.730 | 0.730 | 0.684 | 0.683 |
| | **15K** | 0.730 | 0.729 | 0.691 | 0.691 |
| | **20K** | 0.757 | 0.757 | 0.726 | 0.726 |
| **CITATION SENTIMENT CORPUS (CSC)** | **8K** | 0.872 | 0.863 | 0.838 | 0.842 |
| **BIOINFORMATICS CITATION CORPUS (BCC)** | **4K** | 0.695 | 0.696 | 0.693 | 0.691 |

merely the size of the datasets. We formed training data by picking K instances set to 7K, 10K, 15K, and 20K. On the Hotel Review datasets, the model ROBERTA outperforms all other TL models with an accuracy of 94.57 percent, 94.6 percent, 94.97 percent, and 95.48 percent, respectively. On the Movie Review datasets, the model ELECTRA outperforms all other TL models with an accuracy of 94.36 percent, 93.55 percent, 93.8 percent, and 94.13 percent, respectively. On the SENTIMENT140, the model ROBERTA outperforms all other TL models with an accuracy of 83.64 percent, 82.3 percent, 83.4 percent, and 83.59 percent, respectively. The CSC (CITATION SENTIMENT CORPUS) originally had 7980 citations, with almost 8K citations dataset highly unbalanced. Experiments were conducted by running the models on the entire dataset. BERT outperforms with an accuracy of 93.77 percent. The Bioinformatics Citation Corpus (BCC) originally had 4123 instances. Experiments were conducted by running the models on the entire dataset. BERT outperforms with an accuracy of 92.04percent, indicating the effectiveness of fine-tuning with FARM using adjusted hyperparameters values. The findings of the transformer models and the Universal Language Model Fine-tuning model are displayed, and transformers all-around produce better results than ULMFiT.

In this study, we consider accuracy and F1-score as our primary metrics. The table shows the performance of the proposed model with transfer learning (with-TL). On the Hotel Review datasets, the model ROBERTA outperforms all other TL models with an F1-score of 94.53 percent, 94.56 percent, 94.93 percent, and 95.41 percent, respectively. The ROBERTA model approach outperformed the other three models. On the Movie Review datasets, the model ELECTRA outperforms all other TL models with an F1-score of 94.35 percent, 93.55 percent, 93.8 percent, and 94.12 percent, respectively. The ELECTRA model approach outperformed the other three models. On the SENTIMENT140, the model ROBERTA outperforms all other TL models with an F1-score of 83.64 percent, 82.3 percent, 83.39 percent, and 83.59 percent, respectively. The ROBERTA model approach outperformed the other three models. The CSC (CITATION SENTIMENT CORPUS) originally had 7980 citations, with almost 8K citations dataset highly unbalanced. Experiments were conducted by running the models on the entire dataset. BERT outperforms with an F1-score of 90.22 percent. The Bioinformatics Citation Corpus (BCC) originally had 4123 instances. Experiments were conducted by running the models on the entire dataset. BERT outperforms with an F1-score of 87.98 percent, indicating

the effectiveness of fine-tuning with FARM using adjusted hyperparameters values. The findings of the transformer models and the Universal Language Model Fine-tuning model are displayed, and transformers all-around produce better results than ULMFiT.

A combined perspective of the assessment metrics for datasets is shown in Table 9. Because the datasets are mainly unbalanced, with a majority of Positive and Neutral target labels, the accuracy measure alone cannot be employed as the primary performance assessment metric; instead, the weighted average (F1-score) of precision and recall is used. The table results showed the LSTM model outperformed the CNN model across all datasets. We assembled training data by randomly selecting K instances with sizes of 7K, 10K, 15K, and 20K. On the Hotel Review datasets, the model LSTM surpasses the CNN model with an accuracy of 92.4 percent, 92.5 percent, 92.5 percent, and 94.1 percent, and with F1-Scores of 92.4 percent, 92.4 percent, 92.4 percent, and 94.0 percent, respectively. The CNN model outperforms the LSTM model on the Movie Reviews datasets, with accuracy rates of 83.9 percent, 84.0 percent, 85.7 percent, and 86.7 percent, respectively, and F1-Scores of 83.9 percent, 84.0 percent, 85.6 percent, and 86.6 percent. The LSTM beats the CNN on the SENTIMENT140 Tweet datasets by accuracy rates of 68.8 percent, 73.0 percent, 73.0 percent, and 75.7 percent, F1-Scores of 69.0 percent, 73.0 percent, 72.9 percent, and 75.7 percent, respectively. CITATION SENTIMENT CORPUS (CSC) fared well employing Long Short-Term Memory with an accuracy of 87.2 percent and an F1-Score of 86.3 percent. The accuracy and F1-Score of the Bioinformatics Citation Corpus (BCC) utilizing LSTM are higher at 69.5 percent and 69.6 percent, respectively. CNN also performed admirably across all datasets except LSTM was marginally more promising.

The Table 10 illustrates the comparison of the model with transfer learning (with-TL) and without transfer learning (without-TL) in terms of accuracy. In Fig 6, we could demonstrate that the TL significantly outperform the non-TL model. ROBERTA surpasses all other TL techniques and deep learning models on the Hotel Review datasets, yielding accuracy of 94.57 percent, 94.6 percent, 94.97 percent, and 95.48 percent, correspondingly. LSTM beat CNN among deep learning models. LSTM exceeds CNN because LSTM captures long-term dependencies effectively.

**Table 10. Comparison of TL and DL results based on accuracy.**

| DATASETS | SIZE | Transfer Learning | | | | | Deep Learning |
|---|---|---|---|---|---|---|---|
| | | ULMFIT | BERT | ROBERTA | ELECTRA | LSTM | CNN |
| | | A | A | A | A | A | A |
| HOTEL REVIEW | 7K | 0.928 | 0.941 | **0.946** | 0.918 | 0.924 | 0.901 |
| | 10K | 0.940 | 0.942 | **0.946** | 0.918 | 0.925 | 0.898 |
| | 15K | 0.941 | 0.948 | **0.950** | 0.924 | 0.924 | 0.883 |
| | 20K | 0.952 | 0.947 | **0.955** | 0.931 | 0.941 | 0.904 |
| MOVIE REVIEW | 7K | 0.921 | 0.917 | 0.936 | **0.944** | 0.801 | 0.839 |
| | 10K | 0.921 | 0.908 | **0.941** | 0.936 | 0.820 | 0.840 |
| | 15K | 0.926 | 0.918 | 0.935 | **0.938** | 0.862 | 0.857 |
| | 20K | 0.931 | 0.911 | 0.940 | **0.941** | 0.885 | 0.867 |
| SENTIMENT140 | 7K | 0.754 | 0.804 | **0.836** | 0.795 | 0.688 | 0.656 |
| | 10K | 0.770 | 0.805 | **0.823** | 0.801 | 0.730 | 0.684 |
| | 15K | 0.770 | 0.810 | **0.834** | 0.801 | 0.730 | 0.691 |
| | 20K | 0.780 | 0.814 | **0.836** | 0.812 | 0.757 | 0.726 |
| CITATION SENTIMENT CORPUS (CSC) | 8K | 0.910 | **0.938** | 0.937 | 0.932 | 0.872 | 0.838 |
| BIOINFORMATICS CITATION CORPUS (BCC) | 4K | 0.855 | **0.920** | 0.894 | 0.885 | 0.695 | 0.693 |

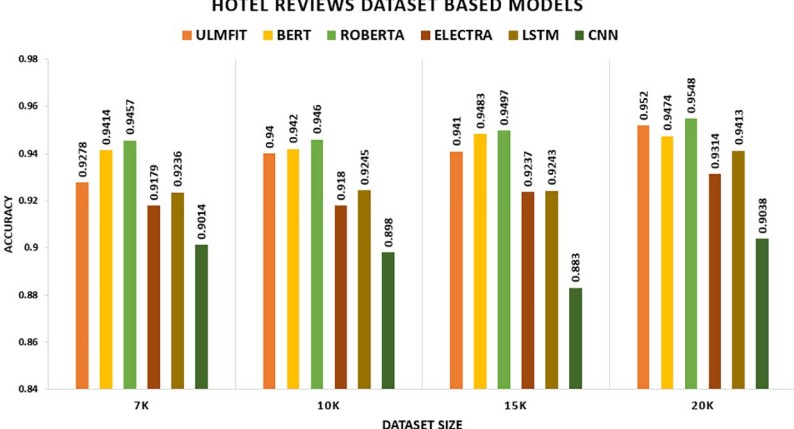

**Fig 6. Comparison graph of hotel reviews TL AND DL results based on accuracy.**

In Fig 7, we were inclined to emphasize that the TL substantially beats the non-TL model in the Movie Review datasets. Initially, the ELECTRA model outperforms all other models with an accuracy of 94.4 percent on 7k, followed by ROBERTA with an accuracy of 94.1 percent on 10k. However, on 15k and 20k, the ELECTRA significantly achieves all other models, with an accuracy of 93.8 percent and 94.1 percent, respectively.

On the SENTIMENT140 datasets, TL significantly outperforms the non-TL model. ROBERTA trumps all other TL methods and deep learning models, yielding accuracy rates of 83.6 percent on 7k, 82.3 percent on 10k, 83.4 percent on 15k, and 83.6 percent on 20k. Among deep learning models, LSTM outscored CNN, achieving accuracy rates of 68.8 percent, 73.0 percent, 73.0 percent, and 75.7 percent, correspondingly shown in Fig 8. In the CITATION SENTIMENT CORPUS, we demonstrate that the TL model surpasses the non-TL model significantly. BERT exceeds all other TL approaches and deep learning models with an accuracy of 93.77 percent, while without TL, LSTM outperforms CNN with an accuracy rate of 87.22 percent shown in Fig 9.

We illustrate that the TL model significantly beats the non-TL model in the Bioinformatics Citation Corpus (BCC). With an accuracy rate of 92.04 percent, BERT surpasses all other TL

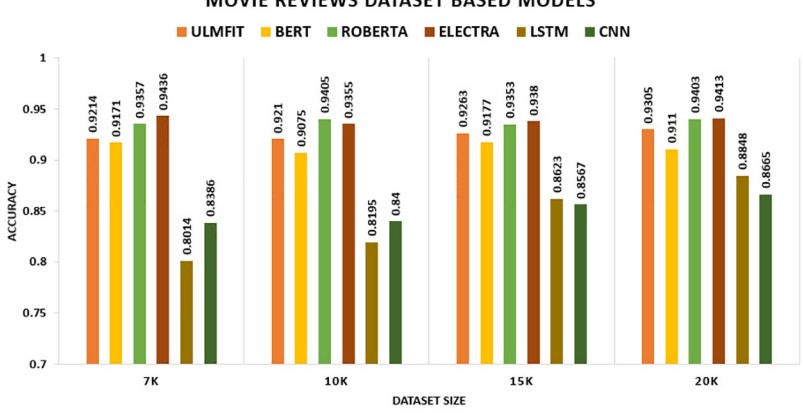

**Fig 7. Comparison graph of movie review TL AND DL results based on accuracy.**

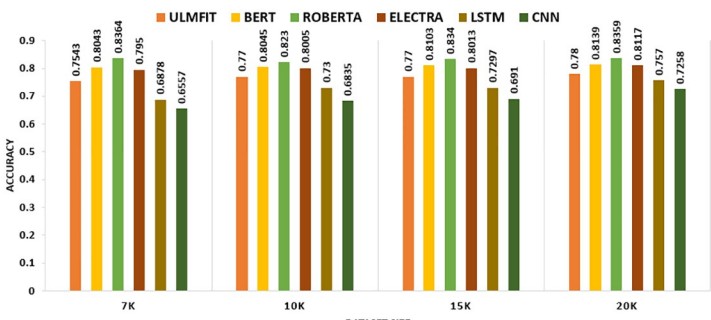

**Fig 8. Comparison graph of Sentiment140 TL AND DL results based on accuracy.**

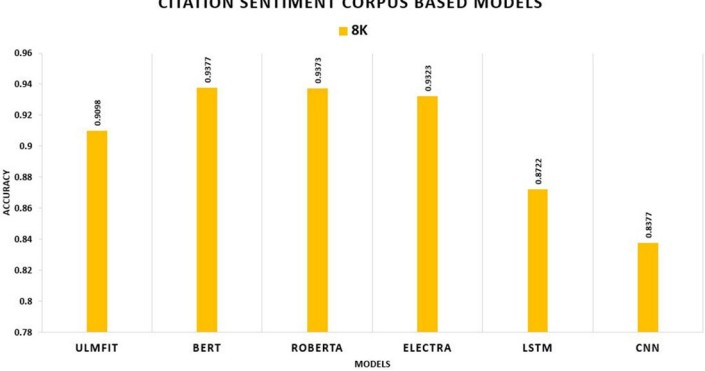

**Fig 9. Comparison graph of CSC TL AND DL results based on accuracy.**

methods and deep learning models, whereas without TL, LSTM outdoes CNN with an accuracy rate of 69.45 percent shown in Fig 10.

The Table 11 illustrates the comparison of the model with-TL and without-TL in terms of F1-SCORE.

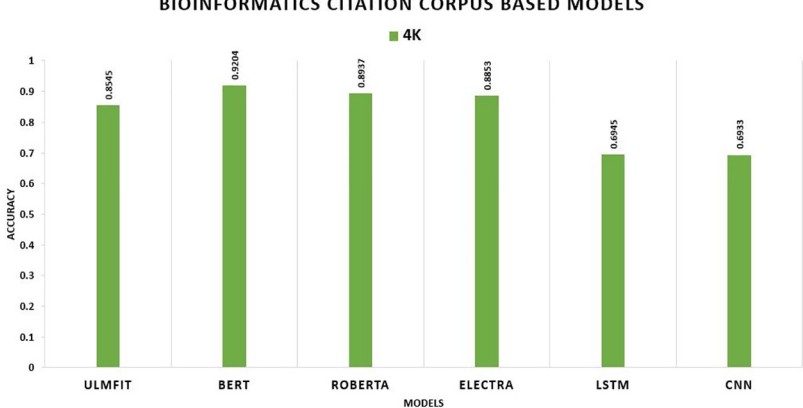

**Fig 10. Comparison graph of BCC TL AND DL results based on accuracy.**

**Table 11. Comparison graph of TL AND DL results based on F1-score.**

| DATASETS | SIZE | Transfer Learning | | | | Deep Learning | |
|---|---|---|---|---|---|---|---|
| | | ULMFIT | BERT | ROBERTA | ELECTRA | LSTM | CNN |
| | | F1 | F1 | F1 | F1 | F1 | F1 |
| HOTEL REVIEW | 7K | 0.927 | 0.941 | **0.945** | 0.917 | 0.924 | 0.901 |
| | 10K | 0.927 | 0.941 | **0.945** | 0.917 | 0.924 | 0.901 |
| | 15K | 0.941 | 0.948 | **0.949** | 0.923 | 0.920 | 0.882 |
| | 20K | 0.951 | 0.947 | **0.954** | 0.930 | 0.940 | 0.903 |
| MOVIE REVIEW | 7K | 0.921 | 0.917 | 0.936 | **0.944** | 0.801 | 0.839 |
| | 10K | 0.921 | 0.908 | **0.941** | 0.936 | 0.820 | 0.840 |
| | 15K | 0.926 | 0.918 | 0.935 | **0.938** | 0.862 | 0.856 |
| | 20K | 0.931 | 0.911 | 0.940 | **0.941** | 0.885 | 0.866 |
| SENTIMENT140 | 7K | 0.754 | 0.804 | **0.836** | 0.795 | 0.690 | 0.654 |
| | 10K | 0.770 | 0.804 | **0.823** | 0.800 | 0.730 | 0.683 |
| | 15K | 0.770 | 0.810 | **0.834** | 0.801 | 0.729 | 0.691 |
| | 20K | 0.780 | 0.814 | **0.836** | 0.812 | 0.757 | 0.726 |
| CITATION SENTIMENT CORPUS (CSC) | 8K | 0.897 | **0.902** | **0.902** | 0.898 | 0.863 | 0.842 |
| BIOINFORMATICS CITATION CORPUS (BCC) | 4K | 0.853 | **0.880** | 0.840 | 0.829 | 0.696 | 0.691 |

We compare TL and non-TL models in our demonstration. The non-TL variant performs noticeably worse than the TL. In the Hotel Review datasets, ROBERTA outpaces all other TL techniques and deep learning models, yielding F1-scores of 94.53 percent, 94.56 percent, 94.93 percent, and 95.41 percent, respectively. In terms of deep learning models, LSTM outdid CNN. By comparison, F1 values obtained without TL are 92.36 percent, 92.36 percent, 92.0 percent, and 94.0 percent shown in Fig 11.

We compare TL and non-TL models in the Movie Review datasets for our study. The TL variant outperforms the non-TL model decisively. Initially, the ELECTRA model surpasses all other models, giving F1-scores of 94.35 percent on the 7k, followed by ROBERTA bettered with F1-scores of 93.55 percent on the 10k. Conversely, the ELECTRA model outperforms all other models on the 15k and 20k scales, attaining F1-scores of 93.8 percent and 94.12 percent, respectively. Regarding deep learning models, Initially, CNN outperformed LSTM with

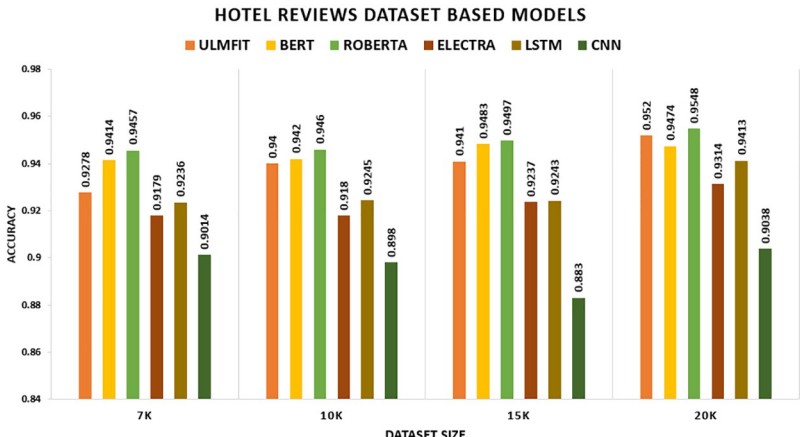

**Fig 11. Comparison graph of hotel reviews TL AND DL results based on F1-score.**

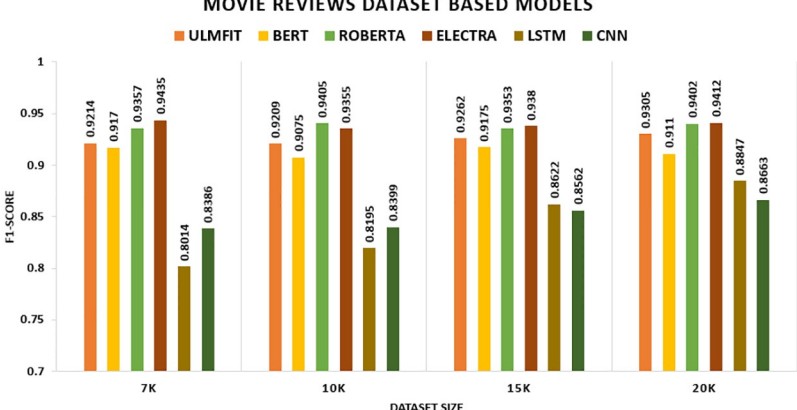

**Fig 12. Comparison graph of movie reviews TL AND DL results based on F1-score.**

F1-scores of 83.9 and 92.36 percent, but LSTM eventually triumphed with 86.2 and 88.5 percent, respectively shown in Fig 12.

On the SENTIMENT140, the model ROBERTA leads all other TL models and deep learning models, with F1 scores of 83.64 percent, 82.3 percent, 83.39 percent, and 83.59 percent, correspondingly. Regarding deep learning models, LSTM outscored CNN, and F1 scores without TL are 69 percent, 73 percent, 72.29 percent, and 75.68 percent, respectively shown in Fig 13.

We assess TL and non-TL models in the CSC. We illustrate that the TL model outperforms the non-TL model by a wide margin. With an F1-score of 90.22 percent, BERT and ROBERTA both outperform all other TL methods and deep learning models. Regarding deep learning models, reaching F1-score of 86.30 percent, LSTM outdoes CNN shown in Fig 14.

We compare TL and non-TL models employing the BCC. The TL variant significantly outperforms the non-TL model. BERT excels all other TL approaches and deep learning models with an F1-score of 87.98 percent, whereas without TL, the F1-score is 69.6 percent shown in Fig 15.

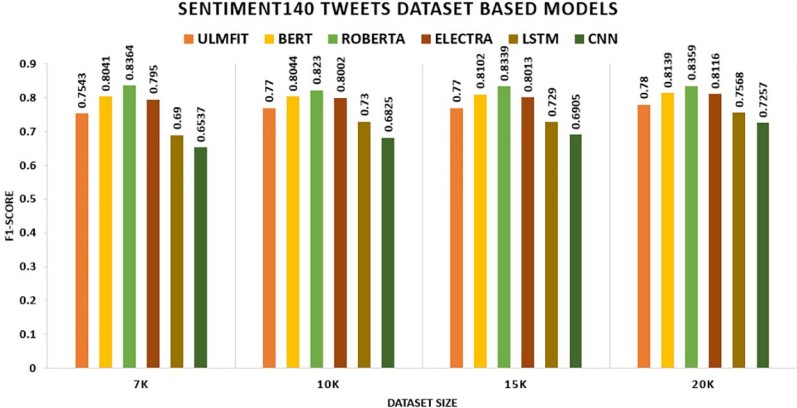

**Fig 13. Comparison graph of Sentiment140 TL AND DL results based on F1-score.**

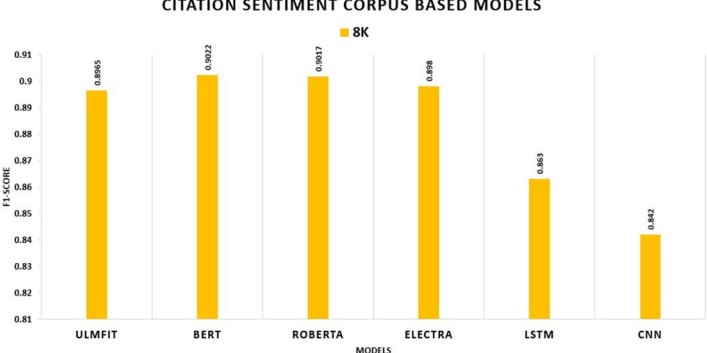

**Fig 14. Comparison graph of CSC TL AND DL results based on F1-score.**

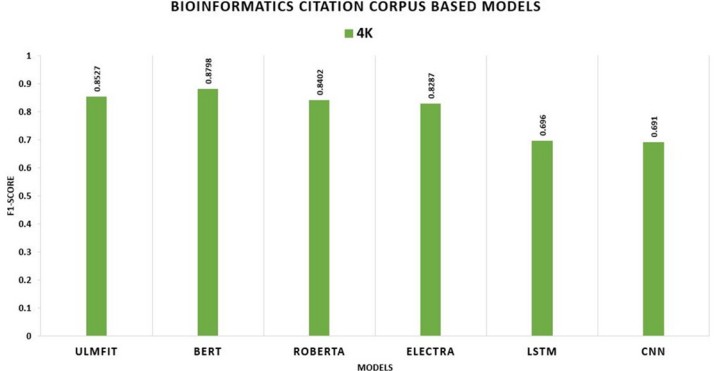

**Fig 15. Comparison graph of BCC TL and DL results based on F1-score.**

## Conclusion

This research demonstrated contemporary transfer learning models in multi-realm sentiment classification. Based on relevant research, we went with techniques that do not necessitate pretraining the whole model from scratch, despite resetting the model's weights using an existing one. It is a practical method for a plethora of reasons: First, when the quantity of accessible textual data is insufficient to meet the requirements for pretraining a completely new model. Second, the computational time, although still significant and depends on the quantity of extra training data, is substantially less than typical pretraining, which would take a week. The primary goal of this study is to address the challenge of limited labeled data for predicting sentiment polarity in a target domain by using a classifier trained on a source domain. This research explores the use of fine-tuned transformers and the highly modular Framework for Adapting Representation Models (FARM) to model sentiment analysis from text. The pretrained models BERT, RoBERTa, and ELECTRA are employed, which are subsequently finetuned with FARM to represent the downstream assessment to identify sentiments, along with ULMFiT. Experiments are conducted on five publicly available sentiment analysis datasets, namely Hotel Reviews (HR), Movie Reviews (MR), Sentiment140 Tweets (ST), Citation Sentiment Corpus (CSC), and Bioinformatics Citation Corpus (BCC). LSTM performs better than CNN in most domains when the results are compared to deep learning models like CNN. FARM framework-based models extensively beat deep learning techniques. Other domain

adaption situations, such as named entity recognition, question answering, and reading comprehension, might benefit from the proposed post-training technique. We would want to examine the use of this idea in these domain adaptation tasks in the future.

## Author Contributions

**Conceptualization:** Imran.

**Data curation:** Sultan Alfarhood.

**Formal analysis:** Tariq Sadad.

**Funding acquisition:** Mejdl Safran.

**Methodology:** Maha Ijaz, Naveed Anwar.

**Supervision:** Naveed Anwar.

**Validation:** Sultan Alfarhood.

**Visualization:** Imran.

**Writing – original draft:** Maha Ijaz.

**Writing – review & editing:** Naveed Anwar, Mejdl Safran, Tariq Sadad.

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
