## [Decision Letter · Decision Letter 0]

1 Sep 2023

PONE-D-23-20322Domain Adaptive Learning for Multi Realm Sentiment Classification on Big DataPLOS ONE

Dear Dr. Sadad,

Thank you for submitting your manuscript to PLOS ONE. After careful consideration, we feel that it has merit but does not fully meet PLOS ONE’s publication criteria as it currently stands. Therefore, we invite you to submit a revised version of the manuscript that addresses the points raised during the review process.

We look forward to receiving your revised manuscript.

Kind regards,

Toqir Rana

Academic Editor

PLOS ONE

2. In your Methods section, please include additional information about your dataset and ensure that you have included a statement specifying whether the collection and analysis method complied with the terms and conditions for the source of the data.

“The authors extend their appreciation to the Deputyship for Research and

Innovation, "Ministry of Education" in Saudi Arabia for funding this

research (IFKSUOR3-013-2)”

“The funders had a role in writing, reviewing, and editing the manuscript”

Reviewers' comments:

Reviewer's Responses to Questions

**Comments to the Author**

1. Is the manuscript technically sound, and do the data support the conclusions?

Reviewer #1: Yes

Reviewer #2: Yes

Reviewer #3: Yes

2. Has the statistical analysis been performed appropriately and rigorously? 

Reviewer #1: No

Reviewer #2: Yes

Reviewer #3: Yes

3. Have the authors made all data underlying the findings in their manuscript fully available?

Reviewer #1: Yes

Reviewer #2: Yes

Reviewer #3: Yes

4. Is the manuscript presented in an intelligible fashion and written in standard English?

Reviewer #1: Yes

Reviewer #2: Yes

Reviewer #3: Yes

5. Review Comments to the Author

Reviewer #1: The paper highlighted the benefits of transfer learning concept for Multi domain Sentimental classification(MDSC)technique on BIG DATA.Transform learning Models like BERT,ULMFiT etc. have been used to study the analysis. Various reliable Sentiment Analysis Datasets were used to influence the outputs.

A thorough Literature Survey of the various transfer learning methods were also done which added a feather to the work done.Analysis using metrics like accuracy & F1 score was done.

Th work can be extended with more relevant metrices and inculsion of statistical analysis will take the wok to a high level.

Reviewer #2: The following comments are required to be incorporated in order to improve the quality of paper and to meet the journal standards.

1 the authors used various Bert based semantic encoding mechanisms, however to make the methods clearer to the readers, it will be better to provide a complete graphical framework of the these methods to illustrate their mechanisms.

2. I did not found any cross validation test for model training.

3. the author should provide complete Hyper-parameters for model training in a tabular form, authors may increase epochs for better training results.

4. The authors should add a model training section in the discussion to explain the effects of the used parameters on the training model by mentioning the following latest predictors such as iHBP-DeepPSSM , cACP-DeepGram, and Deep-AntiFP.

5. The authors should provide an ablution study to visually interpret the proposed model.

Reviewer #3: The proposed idea is interesting, but clarification and corrections are needed.

1.The statistical indicators macro-average F1 score and micro-average F1 score has mentioned in equation 3 and 4 but results were reported of F1 score only. Need to be clarify.

2. write down limitations in current literature.

3. Which split ratio you used for training/testing division?

4. Several typos and grammatical errors are found.

5. Some paragraphs are too long to read.

The authors should try for readability and comprehensibility by dividing paragraphs into two or more.

6.The conclusion section is weak. The conclusion section needs significant revisions. It should briefly describe the findings of the study and some directions for further research.

7. following research papers can be considered in your manuscript for possible references:

a)Hassan, Muhammad Abul, et al. "Intelligent transportation systems in smart city: a systematic survey." 2023 International Conference on Robotics and Automation in Industry (ICRAI). IEEE, 2023.

b)Ullah, Farhat, et al. "A novel approach for emotion detection and sentiment analysis for low resource Urdu language based on CNN-LSTM." Electronics 11.24 (2022): 4096.

c)Hassan, Muhammad Abul, et al. "New advancements in cybersecurity: A comprehensive survey." Big Data Analytics and Computational Intelligence for Cybersecurity (2022): 3-17.

6. PLOS authors have the option to publish the peer review history of their article (what does this mean?). If published, this will include your full peer review and any attached files.

Reviewer #1: No

Reviewer #2: **Yes: **Dr. Hashim Ali

Reviewer #3: No

---

## [Author Response · Author response to Decision Letter 0]

14 Nov 2023

Manuscript ID:  PAPER ID: PONE-D-23-20322

Reviewer Comments, Author Responses and Manuscript Changes

We are very grateful for the reviews provided by the editors and each of the external reviewers of this manuscript titled “Domain Adaptive Learning for Multi Realm Sentiment Classification on Big Data”. The comments are encouraging, and we thank you for giving us the opportunity to resubmit our research work. The suggestions offered by the reviewers have been immensely helpful, and we have responded to them individually, indicating precisely how we addressed each concern or problem and describing the changes we have made. Please see below for our detailed response to the comments. All section numbers refer to the updated manuscript file with tracked changes.

Responses to Reviewer 1

Comment 1: 

The paper highlighted the benefits of transfer learning concept for Multi domain Sentiment classification (MDSC) technique on BIG DATA. Transform learning Models like BERT, ULMFiT etc. have been used to study the analysis. Various reliable Sentiment Analysis Datasets were used to influence the outputs.A thorough Literature Survey of the various transfer learning methods were also done which added a feather to the work done .Analysis using metrics like accuracy & F1 score was done.

The work can be extended with more relevant metrices and inculsion of statistical analysis will take the wok to a high level.

Response: Thank you for the approval. According to the reviewer’s comment, the answer is as follows: 

Thank you for pointing this out. In general, if you are working with an imbalanced dataset where all classes are equally important, using the macro average would be a good choice as it treats all classes equally. If you have an imbalanced dataset but want to assign greater contribution to classes with more examples in the dataset, then the weighted average is preferred. This is because, in weighted averaging, the contribution of each class to the F1 average is weighted by its size. Suppose we have a balanced dataset and want an easily understandable metric for overall performance regardless of the class. In that case, we can go with accuracy, which is essentially our micro F1 score.

Responses to Reviewer 2

Comment 1: 

The authors used various Bert based semantic encoding mechanisms, however to make the methods clearer to the readers, it will be better to provide a complete graphical framework of the these methods to illustrate their mechanisms.

Response: Thank you for bringing this to our attention.

We employed five distinct datasets as well as transfer learning models such as BERT, RoBERTa, ELECTRA, and ULMFiT. We use that Models Graphical Framework to demonstrate its mechanism.

Comment 2: I did not found any cross validation test for model training.

Response:

Following are the reasons for not using cross-validation test with transformers (or deep learning).

Transfer Learning Effectiveness: Transfer learning, especially in the context of pre-trained transformer models like BERT, GPT-3, and their variants, has been remarkably effective in a wide range of natural language understanding tasks. These models are typically pre-trained on massive datasets containing a diverse range of text, making them strong feature extractors. Fine-tuning these pre-trained models on task-specific data often yields excellent results, and this approach reduces the need for extensive cross-validation. With transfer learning, you can effectively leverage pre-trained weights and fine-tune them on your specific task with a relatively small dataset.

Large Datasets: When you have a substantial amount of data, the impact of selecting a bad train/test split becomes less significant. With a small dataset, a poor split could lead to overfitting or under-fitting, making cross-validation crucial for assessing model performance. However, with large datasets, you can often afford to set aside a substantial portion for training and still have a reasonable-sized test set for evaluation.

Comment 3: The author should provide complete Hyper-parameters for model training in a tabular form, authors may increase epochs for better training results.

Response: SECTION: 3.3.4 HYPERPARAMETERS

The Hyper-parameters for each model training with fields Learning Rate, Batch-Size, Epochs and MSL has been mentioned in table 7.

The hyper parameters can be adjusted to produce the best outcomes. These variables need to be manually adjusted because the model itself cannot learn them. The term "hyperparameters" refers to a wide range of variables, such as the learning rate, training batch size, dropout rate, number of training epochs, and any other optimizer-related variables that can be altered, like the Adam optimisation technique. We employ bert-base-cased for BERT [46], roberta-base for RoBERTa [64], and google/electra-base-discriminator for ELECTRA [65] as baselines for deep neural network techniques.

Table 7: HYPERPARAMETERS EMPLOYED FOR BERT, ROBERTA, AND ELECTRA

Responses to Reviewer 3

Comment 1: The statistical indicators macro-average F1 score and micro-average F1 score has mentioned in equation 3 and 4 but results were reported of F1 score only. Need to be clarify.

Response:

We mentioned weighted average F1 score. If you have an imbalanced dataset but want to assign greater contribution to classes with more examples in the dataset, then the weighted average is preferred. This is because, in weighted averaging, the contribution of each class to the F1 average is weighted by its size. Suppose you have a balanced dataset and want an easily understandable metric for overall performance regardless of the class. In that case, you can go with accuracy, which is essentially our micro F1 score.

Comment 2: Write down limitations in current literature.

Response: 

We agree with the reviewer’s assessment. Accordingly, throughout the manuscript, we have revised the literature review according to the current literature and added limitations of the work. 

Comment 3: Which split ratio you used for training/testing division?

Response: 

We’ve used five (5) different datasets and in three of them we used four (4) different samples size for training. And for each sample we have used 70:30 split ratio. 

Comment 4: Several typos and grammatical errors are found.

Response: The grammatical errors have been corrected on [insert the exact location where the change can be found in the revised manuscript].

Comment 5: Some paragraphs are too long to read. The authors should try for readability and comprehensibility by dividing paragraphs into two or more.

Response:

The partition of paragraphs has been corrected accordingly.

Comment 6: The conclusion section is weak. The conclusion section needs significant revisions. It should briefly describe the findings of the study and some directions for further research.

Response:

This research demonstrated contemporary transfer learning models in multi-realm sentiment classification. This research focuses on evaluating the impact of transfer learning on sentiment analysis results and investigating the effectiveness of domain adaptation. The study employs well-known transfer learning models such as BERT, RoBERTa, ELECTRA, and ULMFiT to enhance the performance of sentiment analysis. Various transformer models are used to analyze sentiment, and the performance of LSTM and CNN is compared.

Based on relevant research, we went with techniques that do not necessitate pretraining the whole model from scratch, despite resetting the model’s weights using an existing one. It is a practical method for a plethora of reasons: First, when the quantity of accessible textual data is insufficient to meet the requirements for pretraining a completely new model. Second, the computational time, although still significant and depends on the quantity of extra training data, is substantially less than typical pretraining, which would take a week. The primary goal of this study is to address the challenge of limited labeled data for predicting sentiment polarity in a target domain by using a classifier trained on a source domain. This research explores the use of fine-tuned transformers and the highly modular Framework for Adapting Representation Models (FARM) to model sentiment analysis from text.

The pretrained models BERT, RoBERTa, and ELECTRA are employed, which are subsequently fine-tuned with FARM to represent the downstream assessment to identify sentiments, along with ULMFiT. Experiments are conducted on five publicly available sentiment analysis datasets, namely Hotel Reviews (HR), Movie Reviews (MR), Sentiment140 Tweets (ST), Citation Sentiment Corpus (CSC), and Bioinformatics Citation Corpus (BCC). LSTM performs better than CNN in most domains when the results are compared to deep learning models like CNN. FARM framework-based models extensively beat deep learning techniques. Other domain adaption situations, such as named entity recognition, question answering, and reading comprehension, might benefit from the proposed post-training technique. In future work, we intend to further assess our approach for these domain adaptation challenges and to explore other pretrained models.

Comment 7: Following research papers can be considered in your manuscript for possible references:

Response:

It would have been interesting to explore this aspect. we agree that this is an important consideration in this manuscript.

---

## [Decision Letter · Decision Letter 1]

27 Dec 2023

Domain Adaptive Learning for Multi Realm Sentiment Classification on Big Data

PONE-D-23-20322R1

Dear Dr. Sadad,

We’re pleased to inform you that your manuscript has been judged scientifically suitable for publication and will be formally accepted for publication once it meets all outstanding technical requirements.

Kind regards,

Toqir Rana, Ph.D.

Academic Editor

PLOS ONE

Additional Editor Comments (optional):

Reviewers' comments:

Reviewer's Responses to Questions

**Comments to the Author**

1. If the authors have adequately addressed your comments raised in a previous round of review and you feel that this manuscript is now acceptable for publication, you may indicate that here to bypass the “Comments to the Author” section, enter your conflict of interest statement in the “Confidential to Editor” section, and submit your "Accept" recommendation.

Reviewer #2: All comments have been addressed

Reviewer #3: All comments have been addressed

2. Is the manuscript technically sound, and do the data support the conclusions?

Reviewer #2: Yes

Reviewer #3: Yes

3. Has the statistical analysis been performed appropriately and rigorously? 

Reviewer #2: Yes

Reviewer #3: Yes

4. Have the authors made all data underlying the findings in their manuscript fully available?

Reviewer #2: Yes

Reviewer #3: Yes

5. Is the manuscript presented in an intelligible fashion and written in standard English?

Reviewer #2: Yes

Reviewer #3: Yes

6. Review Comments to the Author

Reviewer #2: The paper is in good shape now. All the technical details are met by the authors. Furthermore, emhasis may olease be made to refine the work in upcoming research

Reviewer #3: Authors in this research article have addressed all the required suggestions given by previous reviewers and i believe this paper is now in the scope of the journal.

7. PLOS authors have the option to publish the peer review history of their article (what does this mean?). If published, this will include your full peer review and any attached files.

Reviewer #2: No

Reviewer #3: No

---

## [Editor Report · Acceptance letter]

30 Jan 2024

PONE-D-23-20322R1 

PLOS ONE

Dear Dr. Sadad, 

I'm pleased to inform you that your manuscript has been deemed suitable for publication in PLOS ONE. Congratulations! Your manuscript is now being handed over to our production team.

Kind regards, 

on behalf of

Dr. Toqir Rana 

Academic Editor

PLOS ONE